# Mechanisms of Action of Sea Cucumber Triterpene Glycosides Cucumarioside A_0_-1 and Djakonovioside A Against Human Triple-Negative Breast Cancer

**DOI:** 10.3390/md22100474

**Published:** 2024-10-17

**Authors:** Ekaterina S. Menchinskaya, Ekaterina A. Chingizova, Evgeny A. Pislyagin, Ekaterina A. Yurchenko, Anna A. Klimovich, Elena. A. Zelepuga, Dmitry L. Aminin, Sergey A. Avilov, Alexandra S. Silchenko

**Affiliations:** 1G.B. Elyakov Pacific Institute of Bioorganic Chemistry, Far Eastern Branch of the Russian Academy of Sciences, Pr. 100-letya Vladivostoka 159, 690022 Vladivostok, Russia; chingizova_ea@piboc.dvo.ru (E.A.C.); pislyagin@hotmail.com (E.A.P.); eyurch@piboc.dvo.ru (E.A.Y.); annaklim_1991@mail.ru (A.A.K.); zel@piboc.dvo.ru (E.A.Z.); daminin@piboc.dvo.ru (D.L.A.); avilov_sa@piboc.dvo.ru (S.A.A.); 2Department of Biomedical Science and Environmental Biology, Kaohsiung Medical University, No. 100, Shin-Chuan 1st Road, Sanmin District, Kaohsiung City 80708, Taiwan

**Keywords:** triterpene glycosides, breast cancer, anticancer activity, molecular mechanisms, mitochondrial apoptotic pathway

## Abstract

Breast cancer is the most prevalent form of cancer in women worldwide. Triple-negative breast cancer is the most unfavorable for patients, but it is also the most sensitive to chemotherapy. Triterpene glycosides from sea cucumbers possess a high therapeutic potential as anticancer agents. This study aimed to identify the pathways triggered and regulated in MDA-MB-231 cells (triple-negative breast cancer cell line) by the glycosides cucumarioside A_0_-1 (Cuc A_0_-1) and djakonovioside A (Dj A), isolated from the sea cucumber *Cucumaria djakonovi*. Using flow cytometry, fluorescence microscopy, immunoblotting, and ELISA, the effects of micromolar concentrations of the compounds on cell cycle arrest, induction of apoptosis, the level of reactive oxygen species (ROS), mitochondrial membrane potential (Δψm), and expression of anti- and pro-apoptotic proteins were investigated. The glycosides caused cell cycle arrest, stimulated an increase in ROS production, and decreased Δψm in MDA-MB-231 cells. The depolarization of the mitochondrial membrane caused by cucumarioside A_0_-1 and djakonovioside A led to an increase in the levels of APAF-1 and cytochrome C. This, in turn, resulted in the activation of caspase-9 and caspase-3 and an increase in the level of their cleaved forms. Glycosides also affected the expression of Bax and Bcl-2 proteins, which are associated with mitochondria-mediated apoptosis in MDA-MB-231 cells. These results indicate that cucumarioside A_0_-1 and djakonovioside A activate the intrinsic apoptotic pathway in triple-negative breast cancer cells. Additionally, it was found that treatment with Cuc A_0_-1 resulted in in vivo inhibition of tumor growth and metastasis of murine solid Ehrlich adenocarcinoma.

## 1. Introduction

A total of 2.3 million cases of female breast cancer (BC) and approximately 0.7 million female deaths were reported in the GLOBOCAN 2020 database for 185 countries or territories in 2020. The growth of newly diagnosed breast cancer cases has been projected [1]. Furthermore, the COVID-19 pandemic has had a significant negative impact on female breast cancer statistics because of limited access to early diagnosis and treatment [2]. The non-surgical treatments for breast cancer include radiotherapy and chemotherapy, comprising endocrine therapy, targeted therapy, and immunotherapy. Some innovative therapies are also being investigated, such as gene therapy, vaccines, and adoptive cell therapies, including T-cell receptor therapy and chimeric antigen receptor T (CAR-T) therapy, and they have achieved promising results [3]. A combination of surgery, radiation therapy, and medications can lead to success. Notably, the specific molecular and cellular features of BC provide advantages for the development of targeted therapies. Nevertheless, the available arsenal of modern drugs for various targeted therapies is insufficient, as evidenced by the high mortality rate of female BC patients, and the search for new antitumor drugs is extremely relevant. More than 50% of existing medicines are based on natural products. Marine invertebrates are a prospective source of secondary metabolites with unique structures and properties that are appropriate for applications in medicinal chemistry and pharmacology [4,5,6,7]. The triterpene glycosides (saponins) of sea cucumbers (class Holothuroidea, phylum Echinodermata) possess diverse biological activities and are used as platforms for the development of immunostimulatory and anticancer drugs. Over the period from 2012 to 2021, 29 in vivo research works and clinical studies of the anticancer properties of saponin/saponin-rich extracts from various holothurians were conducted [8]. Frondoside A, a triterpene glycoside from the sea cucumber *Cucumaria frondosa*, demonstrated antimetastatic activity and suppressed tumor metastasis in vivo [9], and it was reported to be promising for the treatment of bladder cancer [10] and breast cancer [11]. Stichloroside C_2_ from *Thelenota ananas* was found to be a dose-dependent inhibitor of proliferation and colony growth in human and murine triple-negative breast cancer [12]. Despite this, only a pilot phase II clinical trial of a water extract of Malaysian sea cucumber *Stichopus* sp. in patients with untreated asymptomatic myeloma is known [13]. These data clearly indicated that this potentially anticancer natural compound group was underestimated.

Our previous studies showed that cucumarioside A_0_-1 (Cuc A_0_-1) and djakonovioside A (Dj A) (Figure 1) from the sea cucumber *Cucumaria djakonovi* [14] suppressed the proliferation and migration of the triple-negative breast cancer (TNBC) cell line MDA-MB-231 and blocked the growth of tumor cell colonies. Currently, drug design and development would be incomplete without computational chemistry methods. For glycosides isolated from *Cucumaria djakonovi*, a ligand-based drug design approach (or indirect drug design), which is based on the structural information of the active compounds binding to the biological target of interest, was partially applied. The quantitative structure–activity relationships (QSARs) of these triterpene glycosides were studied in terms of their cytotoxic activity against human erythrocytes and structural features [15].

As a continuation of this research, we attempted to decipher the underlying molecular mechanisms of the anticancer action of the glycosides Cuc A_0_-1 and Dj A against MDA-MB-231 cells in vitro. In the present study, we investigated the influence of glycosides on cell cycle progression, apoptosis, ROS and MMP levels, Bax/Bcl-2 ratio, cytochrome C and APAF-1 contents, and activation of caspases in cancer cells. We also assessed the antitumor activity of a more active compound in an in vivo murine model of solid Ehrlich adenocarcinoma.

## 2. Results and Discussion

### 2.1. The Influence of the Glycosides on Cell Cycle Progression

It was previously reported that Cuc A_0_-1 and Dj A significantly inhibited MDA-MB-231 cell migration and blocked colony formation at concentrations below their EC_50_ (0.25–1 μM for Cuc A_0_-1 and 0.5–2 μM for Dj A) [14]. To clarify the mode of their action on cancer cells, the effect of these glycosides on cell cycle progression of MDA-MB-231 cells within 24 h was assessed. Cuc A_0_-1 was tested at concentrations of 0.5 and 1 µM, and Dj A was tested at concentrations of 1 and 2 µM (Figure 2).

At both concentrations, Cuc A_0_-1 increased the percentage of cells in the G_2_/M phase by more than half. The percentage of untreated cells in the G_2_/M phase was about 16%, but it was increased to around 22% and 24% in the cells treated with Cuc A_0_-1 at 0.5 μM and 1 μM, respectively (Table 1).

At the same time, Dj A significantly increased the percentage of S-phase cells to 35% and 42% at concentrations of 1 μM and 2 μM, respectively, as opposed to 31% in the control (Table 1).

Thus, Cuc A_0_-1 arrested the cells in the G_2_/M phase, whereas Dj A prevented the transition from the S- to the G_2_ phase. Cell cycle progression is regulated by the activity of complexes of cyclins with relevant cyclin-dependent kinases (CDKs). The cyclin B/CDK-1 complex controls the progression of the G_2_/M phase, while the transition from the S-phase to the G_2_ phase is controlled by CDK-2, which consistently forms complexes with cyclins E and A. Therefore, the effect of Cuc A_0_-1 on the levels of cyclin B and CDK-1 and the effect of Dj A on the levels of cyclin A and CDK-2 were investigated using the Western blotting technique.

The treatment of MDA-MB-231 cells with Cuc A_0_-1 at concentrations of 0.25, 0.5, and 1 μM caused an increase in cyclin B level of 66–76% (Figure 3a,b) and a decrease in CDK-1 level of 27–33% (Figure 3a,d), while Dj A reduced the level of CDK-2 by 41% and 54% at 1 and 2 μM, respectively (Figure 3a,e), and increased the level of cyclin A by 66%, 143%, 199% dose-dependently at concentrations from 0.5 to 2 μM (Figure 3a,c).

Thus, Cuc A_0_-1 and Dj A arrest the cell cycle in MDA-MB-231 cells by altering cyclins and CDK levels, which can initiate programmed cell death (apoptosis) [16].

Next, the induction of apoptosis in MDA-MB-231 cells after treatment with Cuc A_0_-1 and Dj A was determined.

### 2.2. Induction of Apoptosis in Cells by the Glycosides

The inversion of phosphatidylserine (PS) from the inner monolayer of the cytoplasmic membrane into the outer monolayer is one of the characteristic markers of apoptosis, distinguishing it from necrosis. In cell biomembranes, PS is usually asymmetrically distributed and is present only in the inner lipid monolayer. The appearance of PS on the outer layer of the plasma membrane indicates an early stage of apoptosis [17].

The effect of Cuc A_0_-1 and Dj A on the release of PS to the surface of the cytoplasmic membrane, and therefore, the induction of apoptosis in MDA-MB-231 cells, was assessed using the Annexin V-FITC fluorescent probe and flow cytometry technique. The apoptosis level was measured after 24 h of treatment (Figure 4).

Experiments showed that glycosides induced apoptosis at the studied concentrations. The greatest effect was observed when cells were incubated with Cuc A_0_-1 at a concentration of 1 μM (Figure 4a,b). The percentage of cells at the early stage of apoptosis increased to 56%, while in the untreated control cells, it was 3%. At a concentration of 2 μM, Dj A significantly increased the percentage of apoptotic cells to 11% (Figure 4a,c).

Apoptotic changes in MDA-MB-231 cell nuclei under the action of Cuc A_0_-1 (1 μM) and Dj A (2 μM) for 24 h were detected using Hoechst 33342 staining and fluorescence microscopy. Hoechst 33342 binds to adenine- and thymine-rich regions of the small groove of double-stranded DNA and allows the visualization of chromatin condensation in apoptotic cells. The studied triterpene glycosides led to an increase in dye fluorescence compared to the control, indicating condensation and fragmentation of chromatin in the nuclei (Figure 4d).

Therefore, the induction of apoptosis in MDA-MB-231 cells by Cuc A_0_-1 and Dj A was demonstrated. The next step of the investigation (i.e., the measurement of the level of the signaling molecules involved in apoptosis induction and progression) made it possible to clarify the intracellular pathway of apoptosis induced by the glycosides Cuc A_0_-1 and Dj A.

### 2.3. The Influence of the Glycosides on ROS and MMP Levels

Reactive oxygen species (ROS) are intracellular signaling molecules involved in the activation of the mitochondrial apoptotic pathway [18]. In this regard, the ROS levels in MDA-MB-231 cancer cells treated with Cuc A_0_-1 and Dj A were measured using the fluorescent probe H_2_DCF-DA (Figure 5).

Cuc A_0_-1 at 0.25, 0.5, and 1 μM increased the ROS levels by 28–53% over 6 h (Figure 5a). Dj A at 0.5, 1, and 2 μM increased the ROS levels by 25–72% over 6, 12, and 24 h, respectively (Figure 5b). The most significant effects of both compounds were observed after 6 h. Over 12 and 24 h, there was also a significant increase in ROS; however, over 24 h, the highest fluorescence values at a concentration of 0.5 and 1 μm for Cuc A_0_-1 and Dj A were 46 and 52%, respectively.

The increase in ROS levels causes damage to the mitochondrial membranes and their depolarization. A decrease in mitochondrial membrane potential (MMP) is one of the hallmarks of cells that precedes apoptosis through damage to mitochondrial membranes [18].

The effect of Cuc A_0_-1 and Dj A on mitochondrial membrane potential (MMP) in MDA-MB-231 cells was analyzed using tetramethylrhodamine ethyl ether (TMRE) and JC-1 fluorescent probes, which are accumulated in the mitochondria of living cells in a voltage-dependent manner (Figure 5c–e).

Cuc A_0_-1 at 1 μM decreased the fluorescence of TMRE by 11–35%, and the maximum decrease was detected after 24 h (Figure 5c). Dj A at 1 μM reduced the TMRE fluorescence intensity by 30% after 24 h, while at 2 μM, it diminished MMP by 11%, 12%, and 52% after 6 h, 12 h, and 24 h, respectively (Figure 5d).

JC-1 exists as a green fluorescent monomer on depolarized membranes, whereas when membranes are hyperpolarized, JC-1 forms J-aggregates that are emitted in the orange channel. Changes in the orange/green fluorescence ratio of JC-1 were used to distinguish between healthy and depolarized mitochondria. The influence of glycosides on the orange/green fluorescence ratio of JC-1 was detected using fluorescence microscopy (Figure 5e). Cuc A_0_-1 at 0.5 and 1 μM, as well as Dj A at 1 and 2 μM, strongly increased the green fluorescence (monomeric form) and decreased the orange fluorescence (JC-1 aggregates). These results indicate a decrease in Δψm and depolarization of the mitochondrial membranes (Figure 5e).

The role of ROS in apoptosis is not restricted by their influence on the mitochondrial membrane. The enhanced level of ROS may trigger DNA damage, resulting in cell cycle arrest and apoptotic cell death. In addition, the increase in ROS levels may be the result of p53-dependent repression of SOD genes, which also trigger apoptosis [19].

Thus, the increase in ROS levels in MDA-MB-231 cells under the action of the glycosides Cuc A_0_-1 and Dj A may affect and regulate different intracellular signaling cascades related to tumor progression and proliferation. The influence of glycosides on ROS and MMP in MDA-MB-231 cells can subsequently induce a change in the balance between pro- and anti-apoptotic proteins Bax and Bcl-2 [20]. Therefore, measuring the levels of these proteins is the next stage of research.

### 2.4. The Influence of the Glycosides on the Bax/Bcl-2 Ratio

The effect of Cuc A_0_-1 and Dj A on the levels of cytoplasmic Bax and Bcl-2 proteins in MDA-MB-231 cells over 24 h was investigated via Western blot analysis, where β-actin was used as a background control (Figure 6).

Cuc A_0_-1 at 0.5 and 1 μM increased the Bax levels by 46% and 48%, respectively, and decreased the Bcl-2 levels by 19% and 29%, respectively.

Dj A at 1 and 2 μM caused a more significant increase in Bax levels of 107% and 127%, respectively, and diminished Bcl-2 levels by 30% and 49%, respectively.

Thus, a considerable change in the Bax/Bcl-2 ratio caused by Cuc A_0_-1 and Dj A in MDA-MB-231 cells was observed that confirmed the induction of apoptosis via the mitochondrial (intrinsic) pathway.

### 2.5. Effect of Glycosides on Cyt C and APAF-1 Content

The activation of the pro-apoptotic protein Bax leads to the permeability of the outer membrane of mitochondria and the release of cytochrome C (Cyt C), which binds to the factor activating apoptotic peptidase 1 (APAF-1) and forms an apoptosome [21]. In turn, the apoptosome binds to caspase-9, which subsequently leads to cleavage of caspase-3 [22].

The effects of Cuc A_0_-1 and Dj A on Cyt C and APAF-1 levels in MDA-MB-231 cells were studied via ELISA (Figure 7).

The influence of Cuc A_0_-1 on the Cyt C level in MDA-MB-231 cells exhibited a wave pattern: the concentrations of 0.25 and 0.5 μM increased it by 45% and 23%, respectively, after 6 h. However, this effect was not significant after 12 h of exposure and was again magnified after 24 h of treatment: 0.25, 0.5, and 1 μM enhanced the Cyt C amount by 54%, 34%, and 38%, respectively (Figure 7a).

Dj A in all tested concentrations caused the enhancement of Cyt C amount for all time points (Figure 7b): the dose of 0.5 μM increased it by 30%, 29%, and 54% over 6, 12, and 24 h, respectively; 1 μM of Dj A raised it by 38%, 31%, and 48% over 6, 12, and 24 h, respectively; and 2 μM of Dj A raised it by 41%, 59%, and 33% over 6, 12, and 24 h, respectively.

Moreover, Cuc A_0_-1 at 0.25–1 μM strongly increased the APAF-1 levels by 236–331%, 63–133%, and 13–165% over 6, 12, and 24 h, respectively (Figure 7c), while Dj A at 0.5–2 μM increased the APAF-1 levels by 476–350%, 35–83%, and 61–85% over 6, 12, and 24 h, respectively (Figure 7d).

Thus, both the investigated compounds promoted the release of Cyt C and boosted APAF-1 content, confirming the formation of apoptosomes in TNBC cells [23].

### 2.6. Induction of Caspase Activation Under the Action of the Glycosides

The mitochondrial apoptotic pathway activates downstream caspase-3, which is responsible for execution and cancer cell demise [24]. To determine the involvement of caspase-3/7 in the apoptosis of MDA-MB-231 cells, its activation was determined using the Muse™ Caspase-3/7 Kit and Muse™ Cell Analyzer. The kit contains a DNA dye linked to the DEVD substrate. Cleavage of the substrate by active caspase-3/7 results in dye binding to the DNA and fluorescence. Dead cells were stained with 7-AAD dye. Positive caspase-3/7/negative 7-AAD cells (lower right quadrant) were defined as having active caspase-3/7. Positive caspase-3/7/positive 7-AAD cells (upper right quadrant) were in the late apoptotic/necrotic stage (Figure 8).

Cuc A_0_-1 at 0.5 and 1 µM caused an increase in the percentage of cells with activated caspase-3/7 of 6% and 32%, respectively, over 12 h (Figure 8, Table 2). Dj A at 1 and 2 µM increased this percentage to 4% and 9%, respectively, after 12 h. A more effective activation of caspase-3/7 was observed when cells were incubated with glycosides for 24 h. CucA_0_-1 at 1 µM and Dj A at 2 µM increased the percentage of cells with activated caspase-3/7 to 39% and 18%, respectively, as opposed to 2% in untreated cells (Figure 8, Table 2).

Caspases selectively cleave key proteins, including poly(ADP-ribose) polymerase (PARP-1, nuclear zinc-finger DNA-binding protein). The cleavage of PARP-1 has been extensively used as a marker of apoptosis [25].

The influence of Cuc A_0_-1 and Dj A on the level of cleavage (PARP-1) and caspases-3 and -9 proteins in MDA-MB-231 cells was studied using Western blotting (β-actin was used as a background control (Figure 9)).

Cuc A_0_-1 at a concentration of 0.5–1 μM increased the levels of cleaved caspase-9 by 170% and 205% and cleaved caspase-3 by 130% and 159%, respectively, and also raised the level of cleaved PARP-1 by 35%, 121%, and 122% at the concentrations of 0.25, 0.5, and 1 μM, respectively (Figure 9).

Dj A at concentrations of 0.5–2 μM increased the levels of cleaved caspase-9 by 86–165%, cleaved caspase-3 by 47–96%, as well as cleaved PARP-1 by 39–142%, respectively (Figure 9).

Thus, the obtained data showed that sea cucumber triterpene glycosides Cuc A_0_-1 and Dj A cause cell cycle arrest and trigger the intrinsic pathway of apoptosis in TNBC MDA-MB-231 cells. It was also observed that Cuc A_0_-1 caused these effects in lower concentrations than Dj A. To check the antitumor activity of Cuc A_0_-1, in vivo experiments using the murine breast cancer (Ehrlich carcinoma) model were conducted.

### 2.7. In Vivo Antitumor Activity of Cuc A_0_-1

Cuc A_0_-1 was injected into laboratory mice with transplanted Ehrlich carcinoma in accordance with the scheme described in Section 3.11. Briefly, group I consisted of untreated mice; group II was treated before tumor injection and every second day after; group III received the injection of Cuc A_0_-1 every day; and group IV was treated with doxorubicin as a positive control.

Ehrlich carcinoma cells were stained with PKH800 NIR fluorescent dye before injection into mice (as described in Section 3.11) for lifetime visualization of tumors using the in vivo fluorescence imager, Fluor I IN VIVO. Moreover, tumor metastasis was detected after termination of the experiment when mice were euthanized. PKH800 is a cyanin fluorescent dye with a lipophilic group that binds stably to cell membranes. This dye has maximum fluorescence emission at 797 nm, making it usable for in vivo assays. The area of fluorescence zones in mice inoculated with Ehrlich carcinoma (Figure 10a) and its integrative intensity (Figure 10b) were measured 2, 6, 9, and 12 days after treatment with Cuc A_0_-1. The area of fluorescence zones in untreated mice (group I) increased until day 12 of the observations, while the areas of fluorescence zones in all treated mice (groups II, III, and IV) were significantly reduced. The integrated density of fluorescence zones in untreated mice was significantly decreased as a result of intensive proliferation of tumor cells and a decrease in the amount of fluorescent dye binding to the new generation of cells. At the same time, the integrated density of the fluorescence zones was not reduced in mice in groups II, III, and IV. These in vivo observations indicate a significant antitumor effect of Cuc A_0_-1.

The manual measurement of tumor volume (Figure 10c) confirmed the increase in tumor volume in untreated mice (group I) and its stagnation in groups III and IV. The tumor growth indices at days 9 and 12 of the experiment were calculated for group I as 126% and 201%, respectively (Figure 10d). Cuc A_0_-1, depending on the pattern of use (groups II and III), demonstrated tumor growth inhibition (TGI percentage) of 2% and 22% on day 9 and of 47% and 49% on day 12, respectively. The effect of the glycoside in group III was similar to doxorubicin (group IV), which demonstrated a TGI of 42% and 59% within 9 and 12 days, respectively.

The measurements of mouse body weight during the entire experimental period and tumor weight after termination of the experiment are presented in Table 3.

It was detected that the average tumor weight in untreated mice was 1.21 g when the experiment was completed (Table 3). Cuc A_0_-1 (groups II and III) reduced the average weight of the tumors by 14% and 21%, respectively, and this effect was statistically significant in group III. Similarly, doxorubicin diminished tumor weight by 23%. Additionally, in the group of untreated animals, tumor cachexia expressed in the weakening and weight loss due to depletion of fat and muscle tissue was recorded (Table 3). In group III, the weight loss of the animals was insignificant, and the clinical condition of the mice improved.

Moreover, fluorescence imaging after termination of the experiment detected fluorescence in the areas adjacent to the tumor in mice in the untreated group I, which indicated the metastasis of tumor cells in the abdominal cavity (Figure 11a). In group II, the metastases were significantly smaller than those in group I (Figure 11b), and in groups III and IV, they were not visualized (Figure 11c,d). These observations were in good accordance with earlier in vitro experiments, which showed strong inhibition of cancer cell migration as well as colony formation and growth by Cuc A_0_-1 [14].

Thus, in vivo experiments demonstrated that the sea cucumber glycoside Cuc A_0_-1 inhibited tumor growth and metastasis in mice. This is a great advantage for natural triterpenoids because most of them (especially plant triterpenes) have poor aqueous solubility, limiting their use in cancer therapy [26].

### 2.8. Quantitative Structure–Activity Relationships (QSARs) of Cuc A_0_-1 and Dj A in Relation to Cytotoxicity Against MDA MB-231 Cells

A ligand-based approach was used to determine the relationships between the cytotoxic activity of glycosides against MDA MB-231 cells (pIC_50_) and their structural features (quantitative structure–activity relationships (QSARs)). For this purpose, 246 descriptors characterizing various structural elements, conformational and physicochemical characteristics, topological indices, and functional groups were generated for 20 previously investigated glycosides [14,15] isolated from the sea cucumber *C. djakonovi*. In addition to the descriptors provided by the Calculate Descriptors tool of MOE software 2020.0901 package (polarizability, refractive index, surface charge distribution, dipole moment, hydrogen bonds’ potential strength (donors and acceptors), topological indices describing molecular size and shape, hydrophobic volume, hydrophobic/hydrophilic areas, VdW surface area, atomic valence connectivity index), the following descriptors known to be significant for the activity of the glycosides (the presence/absence of 18(20)-lactone and the side chain, carbohydrate chain branching, and the nature of the second sugar residue, the sulfate groups’ number and positions) were added to the descriptors collection.

Multiple correlation and principal components analyses (PCAs) identified 89 descriptors that influence glycoside cytotoxicity to a certain extent. The validity of the choice of descriptors was supported by the glycosides’ division into two groups (Figure 12).

A linear QSAR model was calculated with the QuaSAR-Model tool and the partial least squares (PLS) and principal component regression (PCR) algorithms of the MOE 2020.0901 CCG software [27] based on the selected descriptors. The best PLS model, comprising 43 meaningful descriptors, fitted well with the experimental data on the cytotoxicity of glycosides and was characterized by good statistical parameters (correlation coefficient r^2^ = 0.9962 and RMSE = 0.0349, cross-validation coefficient r^2^_cros_ = 0.7541 and RMSE_cros_ = 0.2790) (Figure 13).

The comparison of the results of in-silico-based calculations with the observed SAR tendencies showed their conformity. The QSAR results indicated strong positive correlations between the availability of a normal non-shortened side chain, the presence of 18(20)-lactone, and the cytotoxic activity of the glycosides. The negative correlation of molecular volume and shape was confirmed by the higher activity of tetraosides with linear carbohydrate chains than the corresponding pentaosides. The number and positions of sulfate groups were also among the meaningful descriptors but ambiguously affected the activity of the glycosides depending on the architecture of their carbohydrate chains. Thus, the presence of 18(20)-lactone, normal side chain, 16-*O*-acetic, and 23-keto groups in the aglycones, and the sulfate group at C-4 of the first xylose residue, positively affected the cytotoxicity of the tested glycosides against MDA-MB-231 cells. The QSAR data showed a negative correlation between the increase in the quantity of hydroxyls, mainly in the aglycones of the glycosides and in particular positions of their carbohydrate chains, and their cytotoxicity. This was reflected in such descriptors as hydrophobic/hydrophilic areas and distribution, surface area, hydrogen bonds’ potential strength. The finding was confirmed by the weak cytotoxicity of Dj D_1_ against MDA MB-231 cells because of the presence of glucose in its chain as the second residue instead of the quinovose, as well as the branching of the chain with the fifth sugar unit (hydrophobic/hydrophilic areas and charge distribution). The presence of a negatively influencing 23-OH group in the aglycone of Dj A was compensated by the linear tetrasaccharide chain with positively influencing free hydroxyl group at C-2 of the quinovose residue (molecular volume and shape, principal moment of inertia).

Our experiments showed that the tested holothurian triterpene glycosides trigger the intrinsic pathway of apoptosis, increase the level of ROS, and cause a change in Bax/Bcl-2 levels. These effects may probably be connected with upstream signaling cascades launched by the interaction of the glycosides with some membrane receptors. One of the potential membrane targets of these compounds may be the adenosine receptor of A_2B_ subtype, which is known to be overexpressed in MDA-MB-231 cells [28]. This opens further horizons for research on glycosidic anticancer action through the investigation of their interactions with A_2B_AR as the membrane target, both via functional experiments and in silico binding.

## 3. Materials and Methods

### 3.1. Compounds

Cucumarioside A_0_-1 (Cuc A_0_-1) and djakonovioside A (Dj A) were isolated earlier from sea cucumber *Cucumaria djakonovi* [14]. The purity of the compounds was confirmed using ^1^H and ^13^C NMR spectroscopy and mass spectrometry. The compounds were dissolved in ddH_2_O and stored at +4 °C until use.

### 3.2. Cell Lines and Culture Conditions

The human triple-negative breast adenocarcinoma MDA-MD-231 cell line was obtained from the ATCC (HTB-26^™^, Manassas, VA, USA). MDA-MB-231 cells were cultured as monolayers under standard conditions (37 °C, 5% CO_2_) in MEM (Biolot, St. Petersburg, Russia) supplemented with 10% fetal bovine serum (FBS) (Biolot, St. Petersburg, Russia) and 1% penicillin-streptomycin (Biolot, St. Petersburg, Russia).

### 3.3. Analysis of ROS Levels

To assess reactive oxygen species (ROS) levels, MDA-MD-231 cells were seeded at a density of 6 × 10^3^ cells/well in a 96-well plate and incubated for 24 h to allow for adhesion. Following this, the cells were treated with glycosides at different concentrations for 6, 12, and 24 h. To measure ROS generation, 20 µL of a H_2_DCF-DA (Sigma-Aldrich, St. Louis, MO, USA) solution was added to each well, reaching a final concentration of 10.0 µM, and incubated for 30 min at 37 °C in the dark. Fluorescence intensities were recorded using a PHERAstar FS high-speed plate reader (BMG Labtech, Ortenberg, Germany) at excitation and emission wavelengths of 485 nm and 518 nm, respectively. Data were analyzed using MARS Data Analysis software version 3.01R2, with results expressed as percentages relative to the positive control.

### 3.4. Assessment of Mitochondrial Membrane Potential

MDA-MD-231 cells, seeded at a density of 6 × 10^3^ cells per well in a 96-well plate, were treated with varying concentrations of compounds Cuc A_0_-1 or DjA for 6, 12, and 24 h at 37 °C in a 5% CO_2_ incubator. Afterward, each well received a 500 nM solution of tetramethylrhodamine ethyl ester (TMRE) (Lumiprobe RUS Ltd., Moscow, Russian Federation) and was incubated for 30 min at 37 °C. The fluorescence intensity was measured using a PHERAstar FS plate reader (BMG Labtech, Ortenberg, Germany) with excitation at 540 nm and emission at 590 nm. Data were analyzed using MARS Data Analysis v. 3.01R2 (BMG Labtech, Ortenberg, Germany), and the results were expressed as a percentage relative to the control.

MDA-MB-231 cells (3 × 10^4^/mL cells in 12-well plates) were seeded and treated with triterpene glycosides (0.5 and 1 μM for Cuc A_0_-1; 1 and 2 μM for Dj A) for 24 h. The cells were then washed with PBS and stained with LumiTracker^®^ Mito JC-1, according to the manufacturer’s protocol (Lumiprob, Moscow, Russia). The cells were then washed with PBS and observed under a fluorescent microscope MIB-2-FL (LOMO, Moscow, Russia).

### 3.5. Cytochrome C and APAF-1 Determination

MDA-MB-231 cells were seeded in 6-well plates (5 × 10^4^/mL) and incubated for 24 h (37 °C, 5% CO_2_) until complete adhesion was achieved. Glycosides were added to the cells at various concentrations. The cells were incubated with the test substances for 6, 12, and 24 h. Cells incubated without the test compounds were used as controls. RIPA buffer (Sigma-Aldrich, St. Louis, Missouri, USA) was then added to the cells for lysis (10,000× *g*, 15 min, 4 C). The supernatant from the cell lysates was collected and immediately analyzed using cytochrome C (SEA594Hu) and APAF-1 (SEA054Hu) kits using ELISA, according to the manufacturer’s instructions (Cloud-Clone, W. Fernhurst Dr., TX, USA).

### 3.6. *Caspase-3/7* Activation

The assay was performed using the Muse^®^ caspase-3/7 kit, according to the manufacturer’s protocol. Briefly, MDA-MB-231 cells (3 × 10^4^ cells/mL) were seeded into 12-well plates and incubated for 24 h for adhesion. The cells were then treated with triterpene glycosides (0.5 and 1 μM for Cuc A_0_-1; 1 and 2 μM for Dj A) for 12 and 24 h. The cells were then trypsinized and washed with PBS. Cell samples (50 µL) for analysis were prepared in an assay buffer, and 5 µL of caspase-3/7 working solution was added to the cells and incubated at 37 °C for 30 min. Next, 150 μL of the 7-AAD working solution was added to the samples, mixed gently, and analyzed using a Muse^®^ cell analyzer (Luminex, Austin, TX, USA). Data were processed by Muse 1.5 analysis software (Luminex, Austin, TX, USA).

### 3.7. Western Blotting

MDA-MB-231 cells (1 × 10^6^ cells in 5 mL) were seeded in Petri dishes and incubated for 24 h for adhesion. Triterpene glycosides (0.25, 0.5, and 1 μM for Cuc A_0_-1; 0.5, 1, and 2 μM for Dj A) were added and incubated with cells for an additional 24 h. Cells were collected and lysed using RIPA buffer (Sigma-Aldrich, St. Louis, MO, USA). Protein lysates were subjected to electrophoresis in 10–15% SDS-PAGE and transferred to a polyvinylidene fluoride (PVDF) membrane. After transfer, the membrane was blocked in a solution of 5% BSA for 1 h and then incubated with primary antibodies overnight at 4 °C and with secondary antibodies for 1 h at room temperature (Cloud-Clone and Affinity, Wuhan, China). Detection was performed using an ECL solution (Thermo Fisher, Waltham, MA, USA) and ChemiDoc MP imaging system (Bio-Rad, Hercules, CA, USA).

### 3.8. Apoptosis Analysis

The appearance of phosphatidylserine on the outer membrane was examined using flow cytometry and Annexin-V-AF 488/propidium iodide (PI) double staining. Triterpene glycosides were introduced into MDA-MB-231 cells (3 × 10^4^ cells/mL in 12-well plates) at various concentrations and incubated for 24 h. Cells were then collected with trypsin-EDTA solution, stained using the Annexin-V AF 488 kit and propidium iodide from Lumiprobe (Moscow, Russia), and analyzed on a NovoCyte flow cytometer (Agilent, Santa Clara, CA, USA).

### 3.9. Hoechst 33342 Staining

MDA-MB-231 cells (3 × 10^4^/mL cells in 12-well plates) were seeded and treated with the triterpene glycosides (0.5 and 1 μM for Cuc A_0_-1; 1 and 2 μM for Dj A) for 24 h. The cells were then washed with PBS and stained with 5 μg/mL Hoechst 33342 for 15 min at 37 °C. The cells were then washed with PBS and observed under a fluorescence microscope MIB-2-FL (LOMO, Moscow, Russia).

### 3.10. Cell Cycle Analysis

The experiment was performed according to a previously reported method [29]. MDA-MB-231 cells (3 × 10^4^/mL in 12-well plates) were plated for attachment for 24 h. They were treated for 24 h with glycosides (0.5 and 1 μM for Cuc A_0_-1; 1 and 2 μM for Dj A). Next, the cells were collected, washed twice with ice-cold phosphate-buffered saline (PBS), and fixed with 70% ethanol at 4 °C overnight. The cells were then incubated with RNase A for 1 h at 37 °C and stained with propidium iodide solution for 5 min in the dark. Samples were analyzed using a NovoCyte flow cytometer (Agilent, Santa Clara, CA, USA).

### 3.11. In Vivo Experiment

#### 3.11.1. Animals

The study was conducted on mature white female BALB/C mice, weighing 20 ± 1.5 g. The animals were kept under standard conditions in accordance with the rules of SP 2.2.1.3218-14 on the device, equipment, and maintenance of experimental biological vivariums and GOST 33216-2014 “Guidelines for the maintenance and care of animals”. The animals were kept under optimal controlled environmental parameters with a temperature of 23 ± 3 °C, humidity of 50%, and a 12 h light cycle. The mice had constant access to a balanced Delta feed, laboratory animal feed, and filtered water. All experimental work with animals was carried out in accordance with the European Directive 2010/63/EC “On the protection of animals used for scientific purposes” and the rules of GOST 33044-2014 “Principles of good laboratory practice”. Animal experiments were approved by the local ethics committee of the PIBOC FEB RAS, No. 02/24, on 15 April 2024.

#### 3.11.2. Solid Ehrlich Carcinoma

Ehrlich spontaneous mouse breast cancer cells were used for inoculation. First, the cells were stained with the NIR fluorescent dye PKH800 (Lumiprobe, Moscow, Russia) with λem = 797 nm, in accordance with the manufacturer’s instructions. Then, 2 × 10^6^ cells/mouse in 0.2 mL of 0.9% saline solution were injected subcutaneously into the inner thigh of the left hind paw.

The commercial anticancer drug, doxorubicin, was used as the reference drug (positive control). The treatment with Cuc A_0_-1 was carried out with two variants [30,31]. The treatment schemes are presented in Table 4.

Throughout the experiment, changes in tumor size, body weight, and clinical condition of the animals in each group were monitored. Tumor size (average tumor volume) was measured using an electronic caliper in each group of animals. Tumor volume was calculated according to the following formula:V = π/6 × L × W × H,(1)
where V is the volume of the tumor on the day of the experiment, and L, W, and H are the linear dimensions of the tumor node (length, width, and height, respectively).

To assess the dynamics of tumor growth, the tumor growth index (TGI, %) was calculated using the following formula:TGI = (V1 − V0/V0) × 100(2)
where V1 is the volume of the tumor during the control period of observation (9–12 days after tumor inoculation), and V0 is the initial volume of the tumor (day 6 after tumor inoculation).

The fluorescence imaging system “Fluor i In Vivo” (NeoScience, Seoul, Republic of Korea) was used for the visualization of tumors in living mice. Mice were anesthetized with an intramuscular injection of a mixture of Zoletil (Valdepharm, Val-de-Reuil, France) and Rometar (Bioveta, Ivanovice na Hané, Czech Republic) at 25 and 30 mkg/kg, respectively. Anesthetized animals were placed inside the device, and images were captured in the light and NIR channels. Mice without fluorescently labeled inoculated Ehrlich cells were used as the background. Images were processed using NEOimage software (version 1.0.1, NeoScience, Seoul, Republic of Korea).

On the day of termination of the experiment (day 12 after tumor cell injection), the animals were anesthetized with zoletil:rometar (1:4 by volume) and euthanized via cervical dislocation. A layer of tumor tissue was separated to visually assess the condition of the tumor, determine its weight, and calculate the inhibition of tumor growth (ITG, %) in each experimental group. *ITG* calculations were performed according to the following formula:(3)$$ITG=\left(1-\frac{T}{C}\right)\times 100\%$$
where *T* and *C* are the average values of the final tumor weight in the experimental and control groups, respectively.

### 3.12. Building a QSAR Model

The Calculate Descriptor and QuaSAR-Model tools of the MOE 2020.0901 package (Chemical Computing Group, Montreal, QC, Canada) [27] were applied to set up the QSAR model for the set of 20 glycosides isolated from the sea cucumber *C. djakonovi*, for which cytotoxic activity against MDA-MB-231 cells had previously been determined [14,15]. The compounds were divided into two groups, balanced in terms of chemical diversity and activity. A total of 20 conformations for 13 glycosides were used as a training set, and 7 compounds, including Cuc A_0_-1, Dj A, and Dj D_1_, were applied as a test set. To build the QSAR model, the partial least squares (PLS) and principal component regression (PCR) algorithms were used. First, a model was built with the highest correlation by using all descriptors given by the PCR and PLS algorithms; then, unnecessary descriptors had to be eliminated step by step. The procedure included charge calculation and structure optimization, glycoside conformational search and optimization, calculation of a collection of descriptors, correlational analysis, principal component analysis (PCA), removal of descriptors collinear to each other, building and cross-validation of the QSAR model, removal of the descriptors not contributing to the model, and model-checking via creation of a graph showing the correlation between the model-predicted value and the experimental activity value, expressed as pIC_50_. The leave-one-out cross-validation method was applied to the QSAR models obtained to mitigate a possible overfitting risk due to the small size of the dataset and provide a more accurate assessment of our model’s performance. To reduce the number of descriptors, the QSAR model rebuilding procedure was repeated several times to achieve a highly predictive model.

### 3.13. Statistical Analysis

All experiments were performed in triplicate. Data were subjected to statistical analysis using one-way ANOVA tests. Data are shown as the mean ± SEM, and statistical significance was set at *p* ≤ 0.05. All statistical tests were performed using SigmaPlot software (version 14.0; Systat Software Inc., San Jose, CA, USA).

## 4. Conclusions

The current study showed that the treatment of triple-negative breast cancer cells with triterpene glycosides Cuc A_0_-1 and Dj A, isolated from the sea cucumber *Cucumaria djakonovi*, arrests the cell cycle in diverse ways. Cuc A_0_-1 significantly increased the number of cells in the mitotic phase (G_2_/M) by affecting the expression of cyclin B and CDK-1, whereas Dj A arrested the cell cycle in the S-phase and modified the levels of cyclin A and CDK-2. In addition, the glycosides were shown to increase ROS levels in MDA-MB-231 cells, followed by a decrease in the membrane potential of mitochondria, which led to enhanced expression of the pro-apoptotic protein Bax and a decrease in the level of the apoptosis inhibitor protein Bcl-2. Moreover, the glycosides caused an inversion of PS on the outer cytoplasmic membrane, chromatin condensation, and fragmentation, as well as activation of the apoptotic markers caspase-3/7, the cleaved forms of caspases-3, -9, and PARP-1. The obtained data showed that Cuc A_0_-1 and Dj A induced apoptosis in MDA-MB-231 cells via the mitochondrial pathway. Additionally, it was demonstrated that treatment of mice with Ehrlich carcinoma using Cuc A_0_-1 leads to the inhibition of tumor growth and metastasis in these organisms. Thus, a series of data were obtained, proving that sea cucumber triterpene glycosides are excellent leading compounds for the treatment of triple-negative breast cancer.

## Figures and Tables

**Figure 1 marinedrugs-22-00474-f001:**
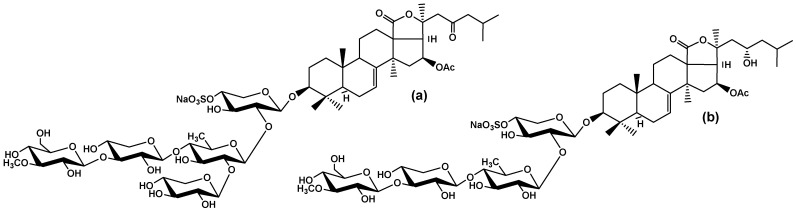
Chemical structures of triterpene glycosides: cucumarioside A_0_-1 (**a**) and djakonovioside A (**b**) isolated from the sea cucumber *Cucumaria djakonovi*.

**Figure 2 marinedrugs-22-00474-f002:**
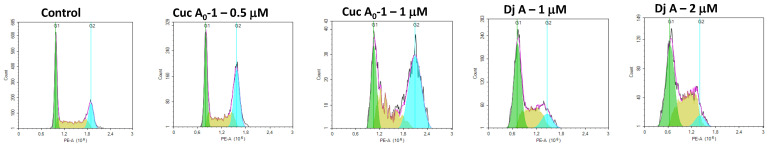
Distribution of MDA-MB-231 cells according to the phases of the cell cycle after treatment with various concentrations of Cuc A_0_-1 and Dj A for 24 h.

**Figure 3 marinedrugs-22-00474-f003:**
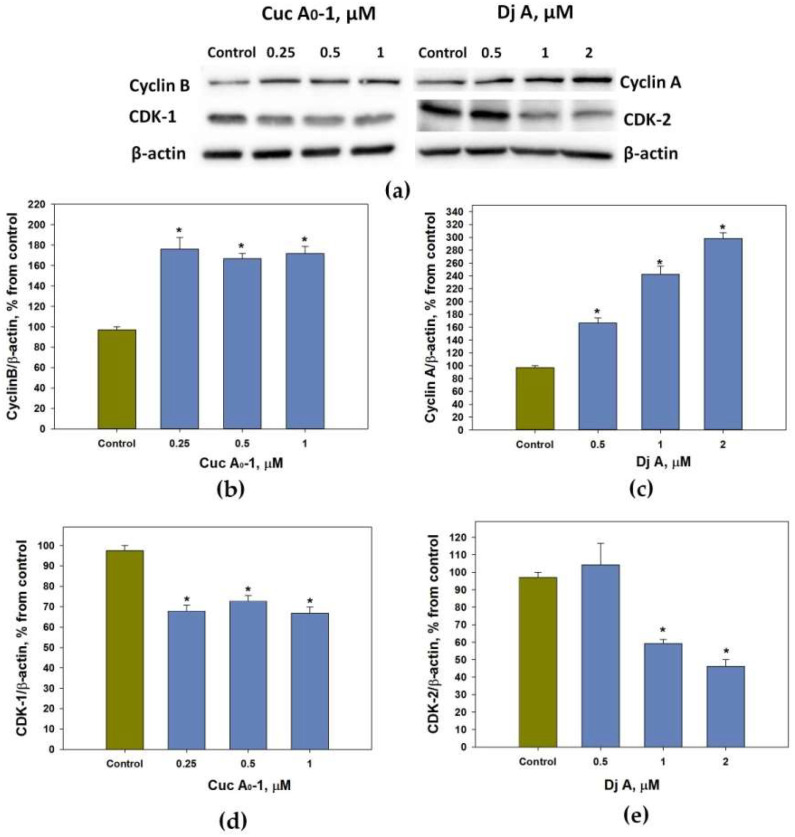
Visualization of cyclin B and A contents and cyclin-dependent kinases in MDA-MB-231 cells treated with triterpene glycosides Cuc A_0_-1 and Dj A at different concentrations. Representative Western blot membranes showing the effect of glycosides on cyclin and CDK protein expression levels (**a**). Processed data on cyclin B content in MDA-MB-231 cells treated with Cuc A_0_-1 (**b**). Processed data on cyclin A content in MDA-MB-231 cells treated with Dj A (**c**). Processed data on CDK-1 content in MDA-MB-231 cells treated with Cuc A_0_-1 (**d**). Processed data on CDK-2 content in MDA-MB-231 cells treated with Dj A (**e**). All data were normalized to the β-actin levels. Data are presented as means ± SEM. * *p* value < 0.05 was considered significant.

**Figure 4 marinedrugs-22-00474-f004:**
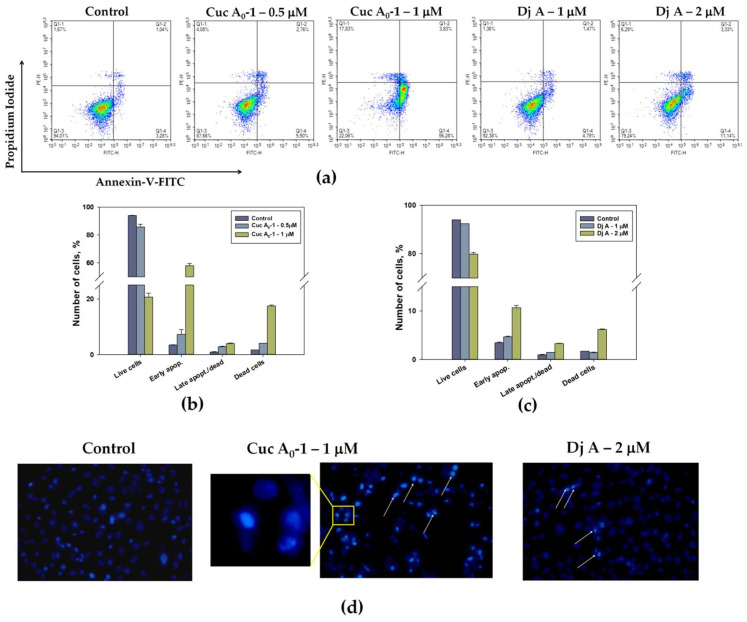
Analysis of apoptosis induced by triterpene glycosides in MDA-MB-231 cells after 24 h of incubation. Flow cytometry assay for Annexin V-FITC/PI staining (**a**). Quantitative calculation of the data obtained via flow cytometry: Cuc A_0_-1 (0.5 and 1 μM)—(**b**) and Dj A (1 and 2 μM)—(**c**). Data are presented as means ± SEM. *p* value < 0.05 was considered significant. Apoptosis assay using Hoechst 33342 in a fluorescent microscopy analysis (**d**). Hoechst 33342 staining showed an increase in chromatin condensation and DNA fragmentation in apoptotic cells treated with Cuc A_0_-1 (1 μM) and Dj A (2 μM) compared with untreated control cells. Arrows indicate nuclei with condensed chromatin.

**Figure 5 marinedrugs-22-00474-f005:**
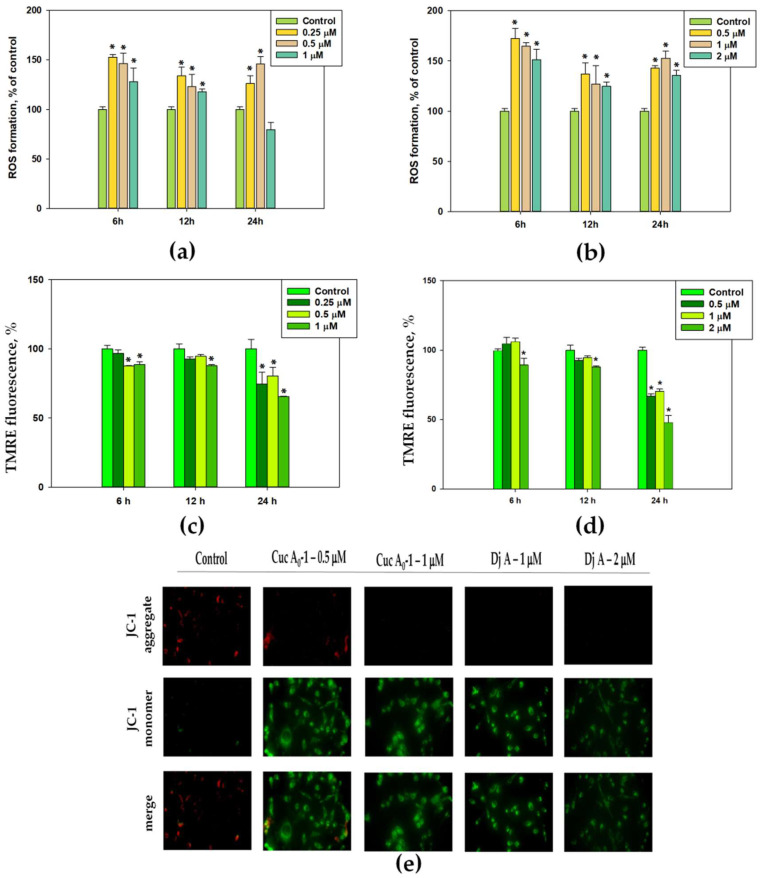
Quantitative evaluation of ROS levels in MDA-MB-231 cells after incubation with Cuc A_0_-1 (**a**) and Dj A (**b**) for different times (6, 12, and 24 h) using the fluorescent dye H_2_DCF-DA. Glycosides Cuc A_0_-1 (**c**) and Dj A (**d**), at various concentrations, reduced the mitochondrial membrane potential (Δψm), as measured using the fluorescent dye TMRE. Data are presented as means ± SEM. * *p* value < 0.05 was considered significant. Staining of MDA-MB-231 cells with the fluorescent dye JC-1 showed a change in the mitochondrial membrane potential (**e**).

**Figure 6 marinedrugs-22-00474-f006:**
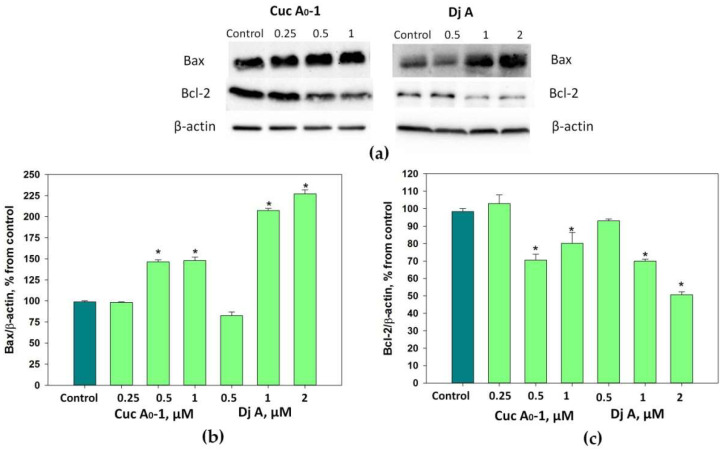
Western blot analysis of cytoplasmic proteins: apoptosis promoter Bax (**a**,**b**) and apoptosis inhibitor Bcl-2 (**a**,**c**) with β-actin as a protein loading control under the treatment of MDA-MB-231 cells with different concentrations of Cuc A_0_-1 and Dj A. Cytoplasmic protein levels were normalized to the control group (untreated cells). * *p* < 0.05 compared with untreated MDA-MB-231 cells.

**Figure 7 marinedrugs-22-00474-f007:**
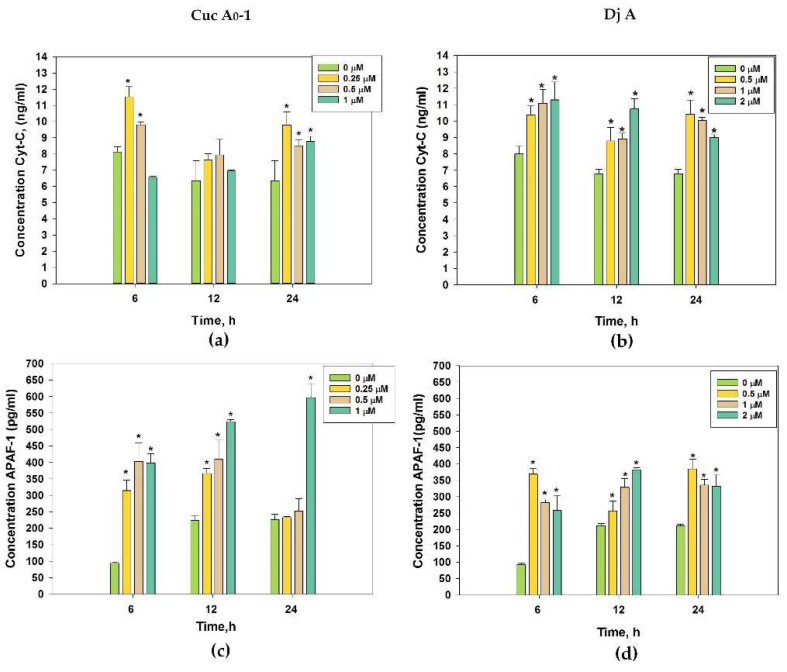
Quantitative assessment of the contents of cytochrome C (**a**,**b**) and APAF-1 (**c**,**d**) in MDA-MB-231 cells after treatment with different concentrations of glycosides Cuc A_0_-1 (**a**,**c**) and Dj A (**b**,**d**) at different times (6, 12, and 24 h) using ELISA kits. * *p* < 0.05 compared with untreated MDA-MB-231 cells.

**Figure 8 marinedrugs-22-00474-f008:**
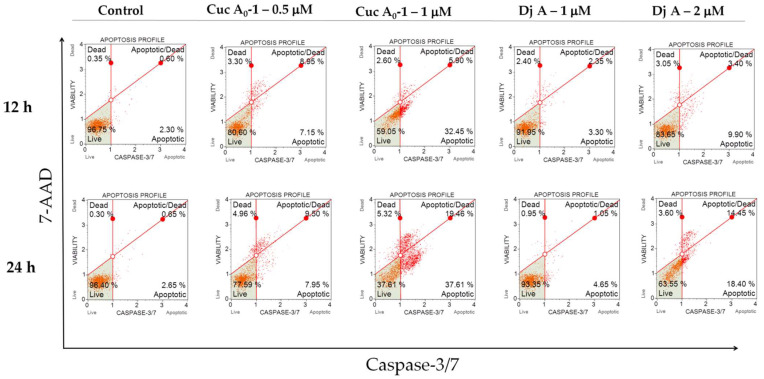
Caspase-3/7 activity in the control cells and cells treated with triterpene glycosides for 12 and 24 h was measured using the Muse™ Caspase-3/7 Kit and flow cytometry in MDA-MB-231 cells.

**Figure 9 marinedrugs-22-00474-f009:**
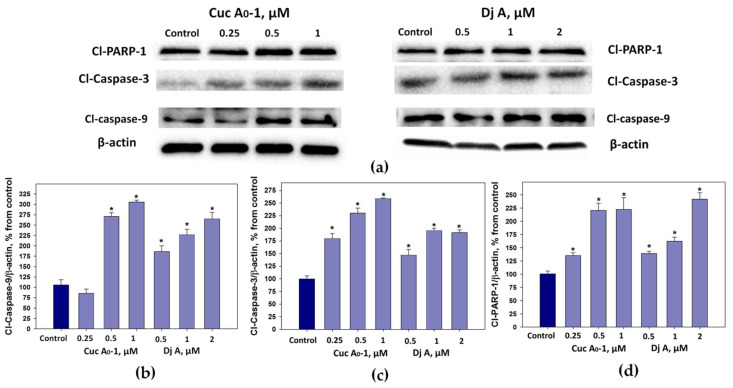
Western blot analysis of apoptotic markers (**a**) and quantitative analysis of the levels of cleaved caspase-9 (**b**), cleaved caspase-3 (**c**), and cleaved PARP-1 (**d**) in MDA-MB-231 cells treated with different concentrations of Cuc A_0_-1 and Dj A. β-actin was used as a protein loading control (**a**). The levels of apoptotic markers were normalized to those of the control group (untreated cells). * *p* < 0.05 compared to untreated MDA-MB-231 cells.

**Figure 10 marinedrugs-22-00474-f010:**
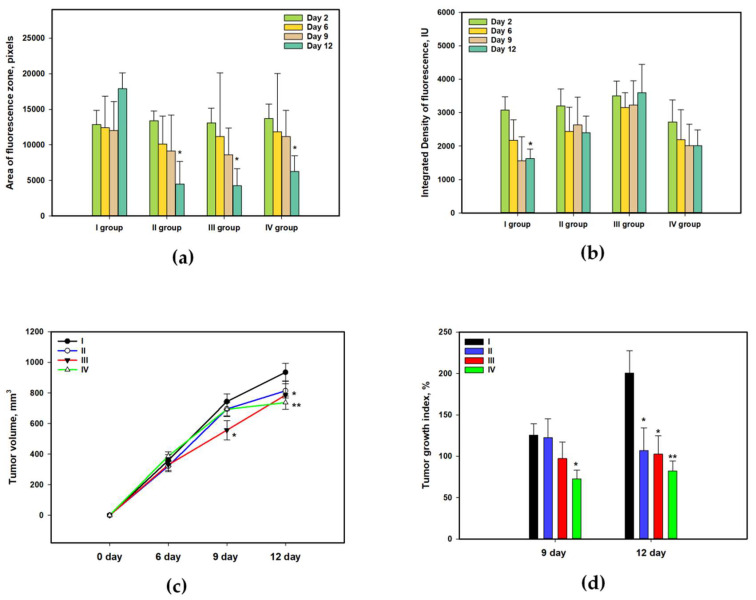
Influence of Cuc A_0_-1 on the area (**a**) and integrated density (**b**) of the fluorescence zone detected by in vivo fluorescence imager, Fluor I. Effect of Cuc A_0_-1 (0.4 µg/mL) on tumor volume (**c**) and tumor growth index (**d**). The data are presented as a mean ± SEM (n = 7). Asterisks indicate the significance of the differences at *p* ≤ 0.05 * and *p* ≤ 0.01 ** according to one-factor analysis of variance (ANOVA) with Tukey’s correction.

**Figure 11 marinedrugs-22-00474-f011:**
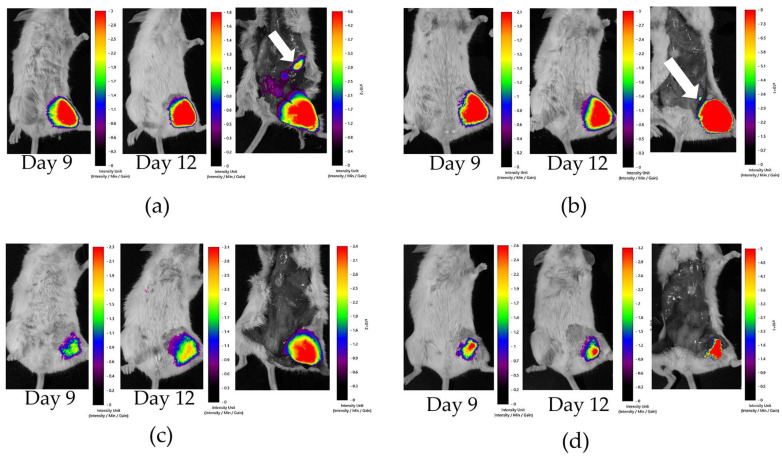
The visualization of tumor cells labeled with PKH800 NIR fluorescent dye using the fluorescence imager system, “Fluor I IN VIVO”, in untreated mice (**a**), mice treated with Cuc A_0_-1 in group II (**b**), mice treated with Cuc A_0_-1 in group III (**c**), and mice treated with doxorubicin in group IV (**d**). On day 12, the tumor area was visualized in live mice; afterward, the mice were euthanized, the skin was opened, and tumor cells were visualized again. Arrows indicate tumor metastasis in the abdominal cavity.

**Figure 12 marinedrugs-22-00474-f012:**
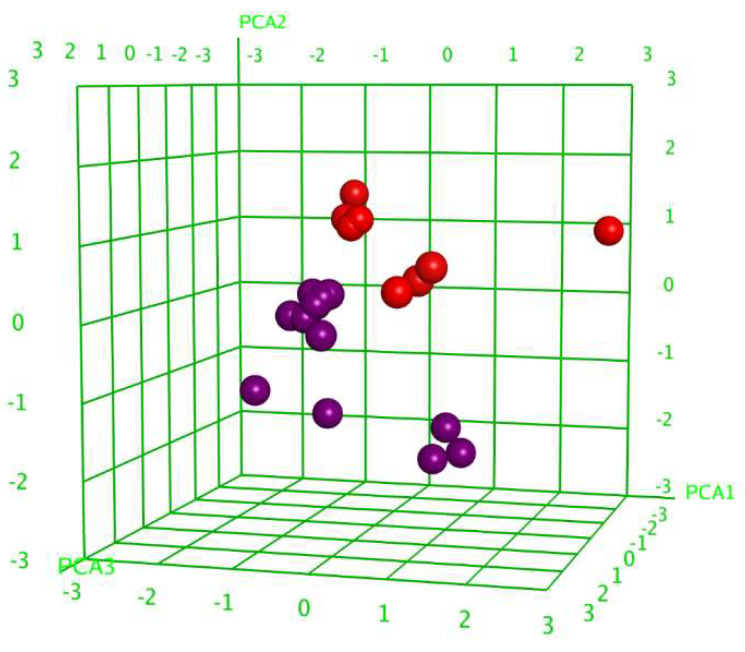
Three-dimensional plot of cytotoxic activity (pIC_50_) dependence on the principal component values (PCA1–PCA3) calculated for 25 conformational forms of 20 glycosides tested against MDA-MB-231 cells. The glycosides demonstrating cytotoxic activity with IC_50_ ≤ 10 μM were outlined as active and are marked in red, while inactive glycosides are marked in violet.

**Figure 13 marinedrugs-22-00474-f013:**
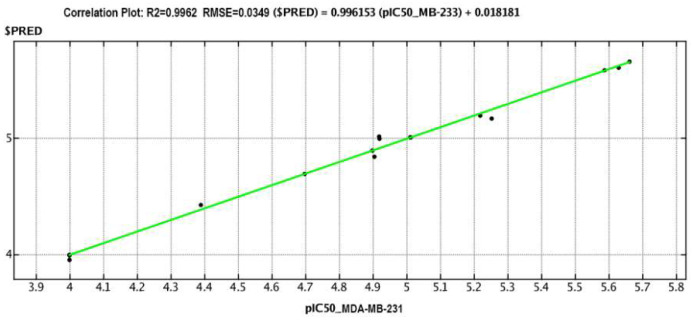
The PLS QSAR model correlation plot reflecting the relationship of predicted and experimental cytotoxicity of the glycosides against MDA-MB-231 cells. The cytotoxic action was expressed as pIC_50_.

**Table 1 marinedrugs-22-00474-t001:** Effects of Cuc A0-1 and Dj A on cell cycle progression of MDA-MB-231 cells within 24 h.

	Cell Cycle Phases (%)		
	G_0_/G_1_	S	G_2_/M
		24 h	
Control	53.36 ± 1.02	31.06 ± 1.46	15.6 ± 0.44
Cuc A_0_-1 (0.5 μM)	45.27 ± 1.98	31.82 ± 0.62	22.32 ± 3.21 *
Cuc A_0_-1 (1 μM)	46.32 ± 1.19	29.24 ± 0.78	24.45 ± 0.42 *
Dj A (1 μM)	52.83 ± 0.52	35.14 ± 0.98 *	12.05 ± 1.51
Dj A (2 μM)	47.33 ± 1.18	42.88 ± 0.73 *	8.01 ± 0.16

* *p* ≤ 0.05 compared with the control.

**Table 2 marinedrugs-22-00474-t002:** Influence of triterpene glycosides on caspase 3/7 activity in MDA-MB-231 cells. The incubation times of cells with the compounds were 12 and 24 h.

	Caspase-3/7 -/7-AAD -	Caspase-3/7 +/7-AAD -	Caspase-3/7 +/7-AAD-+	Caspase-3/7 -/7-AAD +
12 h				
Control	97.50 ± 0.20	1.68 ± 0.28	0.63 ± 0.08	0.20 ± 0.02
Cuc A_0_-1 (0.5 μM)	81.57 ± 0.98	6.12 ± 0.97 *	8.93 ± 0.03 *	3.33 ± 0.24
Cuc A_0_-1 (1 μM)	59.22 ± 0.18	31.65 ± 0.80 *	6.23 ± 0.63 *	2.60 ± 0.12
Dj A (1 μM)	91.53 ± 0.43	3.58 ± 0.28	2.90 ± 0.55	2.00 ± 0.40
Dj A (2 μM)	85.40 ± 1.75	8.80 ± 1.10 *	3.13 ± 0.28	2.68 ± 0.37
24 h				
Control	96.58 ± 0.17	2.48 ± 0.17	0.63 ± 0.03	0.33 ± 0.03
Cuc A_0_-1 (0.5 μM)	76.44 ± 1.15	7.40 ± 0.55 *	10.53 ± 1.03 *	5.63 ± 0.67
Cuc A_0_-1 (1 μM)	40.48 ± 2.87	39.18 ± 1.57 *	16.91 ± 2.56 *	3.44 ± 1.87
Dj A (1 μM)	92.90 ± 0.45	4.65 ± 0.06 *	1.05 ± 0.11	1.18 ± 0.23
Dj A (2 μM)	61.83 ± 1.72	18.48 ± 0.08 *	15.33 ± 0.87 *	4.38 ± 0.78

* *p* ≤ 0.05 compared with control.

**Table 3 marinedrugs-22-00474-t003:** Effect of Cuc A_0_-1 on mouse body and tumor weights.

Group	Body Weight, g				Tumor Weight, g	TGI, %
	Before Inoculation	Day 6	Day 9	Day 12	Day 12	
I	19.18 ± 2.40	20.55 ± 2.53	19.41 ± 1.46	18.29 ± 1.44	1.21 ± 0.14	-
II	19.03 ± 0.54	21.67 ± 0.41	19.94 ± 0.34	18.71 ± 0.41	1.01± 0.15	13.5
III	20.53 ± 1.02	22.06 ± 1.06	21.72 ± 0.48	20.47 ± 1.03 *	0.96± 0.16 *	20.5 *
IV	21.09 ± 1.25	23.47 ± 1.47	21.40 ± 0.71	20.13 ± 1.42 *	0.94 ± 0.07 *	22.5 *

Body weight was measured on days 6, 9, and 12 after inoculation with tumor cells. The data are presented as a mean ± SEM (n = 7). The weight of the tumor tissue was measured 12 days after tumor inoculation. The data presented as the mean ± SEM (n = 7) * indicate that the difference in comparison with group I is significant at *p* ≤ 0.05 according to one-factor analysis of variance (ANOVA) with Tukey’s correction. TGI is the percentage of tumor growth inhibition.

**Table 4 marinedrugs-22-00474-t004:** Treatment scheme for Ehrlich ascites carcinoma mice.

Group	Compound	Start	Treatment	Start	Frequency
I	Control	1 day after cancer cell inoculation	0.9% saline solution	0.5 mL	Daily
II	Cuc A_0_-1	1 day after cancer cell inoculation	0.4 µg/mL	0.5 mL	Every second day
III	Cuc A_0_-1 intensive	6 days after cancer cell inoculation	0.4 µg/mL	0.5 mL	Daily
IV	Doxorubicin	6 days after cancer cell inoculation	0.4 µg/mL	0.5 mL	Daily

## Data Availability

The original data are available from the corresponding author upon request.

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
