# Peer review of "Mechanisms of Action of Sea Cucumber Triterpene Glycosides Cucumarioside A0-1 and Djakonovioside A Against Human Triple-Negative Breast Cancer"

_marinedrugs, 2024, doi:10.3390/md22100474_

Round 1
Reviewer 1 Report
Comments and Suggestions for Authors
Triterpene glycosides cucumarioside A0-1 and djakonovioside A from from the sea cucumber Cucumaria djakonovi possess significant anticancer action, especially human triple negative breast cancer. In order to study molecular mechanism of two compounds on human triple breast cancer, this study evaluated the effect and mechanism of them on human triple negative breast cancer by flow cytometry, fluorescence microscopy, immunoblotting and ELISA, the effect of micromolar concentrations of the compounds on the cell cycle arrest, induction of apoptosis, the level of reactive oxygen species (ROS), mitochondrial membrane potential (Δψm) and the expression of anti- and pro-apoptotic proteins in vivo and in vitro, and providing valuable insights for molecular mechanism study of cucumarioside A0-1 and djakonovioside A.
Although the results obtained in this article were carried out in the laboratory, the research results still have a reference and application value. Meanwhile, there are some points that the authors may need to deal with.
Q1. Moderate English language and style changes required.
Q2. In this paper, corresponding positive controls should required to make the experimental results more convincing?
Q3. Please authors confirm what meaning expressed is the live and dead cells labeled in Figure 4 (b).
Q4. Please explain the optimal concentration for inhibiting the production of ROS and whether there is a concentration dependent inhibition of ROS in fig. 5.
Q5. Could ddH2O effectively dissolve compounds? If it cannot effectively dissolve, how to correctly express the concentration in line 612?
Q6. In the experiment, please accurately indicate the ?-well plate used in line 621-622. Accurately.
Q7. Please confirm “(1 × 106 cells/well )” if author should use “well” in line 666.

Comments on the Quality of English Language
Moderate English language and style changes required.
Author Response
We are very grateful for the time and efforts you spent to evaluate our manuscript.
Q1: Moderate English language and style changes required.
A: Thank you for your careful review of our work. We have made corrections in English.
Q2. In this paper, corresponding positive controls should required to make the experimental results more convincing?
A: We thank the reviewer for the comment. In our previous work, when studying the cytotoxic activity of glycosides, we used cisplatin as a positive control (doi: 10.3390/ijms241311128). We found that the concentrations of glycosides are an order of magnitude lower than the concentration of cisplatin. In this work, we assessed the effect of glycosides on cells in comparison with untreated cells. This approach is used in studying the anticancer activity of compounds (doi: 10.3390/ijms242115922; doi: 10.3390/molecules25184254). At the same time, when studying the antitumor activity in animals, we used doxorubicin as a positive control.
Q3. Please authors confirm what meaning expressed is the live and dead cells labeled in Figure 4 (b).
A: Each dot-plot has 4 quadrant markers representing different cellular states: the upper left quadrant contains dead cells (necrosis) FITC (-)/PI (+), the upper right quadrant contains late apoptotic/dead cells FITC (+)/PI (+), the lower left one contains live cells FITC (-)/PI (-), and the lower right one contains early apoptotic cells FITC (+)/PI (-). Based on the dot-plots in Figure 4a, the graphs in Figures 4b and 4c were constructed. The graphs show the total number of cells.
Q4. Please explain the optimal concentration for inhibiting the production of ROS and whether there is a concentration dependent inhibition of ROS in fig. 5.
А: All studied concentrations of glycosides showed similar activity with respect to ROS. At 6 and 12 h of incubation, dose dependence was observed and the minimal concentrations of substances showed a greater effect. At the same time, at 24 h of incubation, Cuc A0-1 at a concentration of 0.5 μM and Dj A at a concentration of 1 μM showed the greatest ability to cause ROS formation. We have added a description of this data to the results.
Q5. Could ddH2O effectively dissolve compounds? If it cannot effectively dissolve, how to correctly express the concentration in line 612?
A: Triterpene glycosides are highly soluble in ddH2O because they are hydrophilic molecules due to the presence of carbohydrate chains and sulfate groups.
Q6. In the experiment, please accurately indicate the ?-well plate used in line 621-622. Accurately.
А: We apologize for the inaccurate description of the method. These experiments used 96-well plates.
Q7. Please confirm “(1 × 106cells/well )” if author should use “well” in line 666.
А: Thank you for your comment. We made a mistake, the cell concentration was 1 × 106 cells per 5 ml in Petri dishes. We have corrected this in the manuscript text.
Reviewer 2 Report
Comments and Suggestions for Authors
Journal: Marine Drugs
Manuscript ID: marinedrugs-3187184
Manuscript Type: Article
Title: Molecular mechanisms of anticancer action of sea cucumber triterpene glycosides cucumarioside A0-1 and djakonovioside A against human triple negative breast cancer
Authors: Ekaterina S. Menchinskaya, Ekaterina A. Chingizova, Evgeny A. Pislyagin, Ekaterina A. Yurchenko, Anna A. Klimovich, Elena. A. Zelepuga, Dmitry L. Aminin, Sergey A. Avilov, Alexandra S. Silchenko
This study reports identification of the molecular targets and the signaling pathway for the cytotoxicity against MDA-MB-231 human triple negative breast cancer (TNBC) cells observed for the triterpene glycosides, cucumarioside A0-1 (Cuc A0-1) and djakonovioside A (Dj A), isolated from the sea cucumber Cucumaria djakonovi. In MDA-MB-231 cells, Cuc A0-1 and Dj A stimulated ROS production and decreased mitochondrial membrane potential (Δψm) to lead to an increase in the level of APAF-1 and cytochrome C, which resulted in the activation of caspase-9 and caspase-3.These have been supported by the molecular docking profiles for these compounds and the adenosine receptor A2B subtype (A2BAR) that is overexpressed on the membrane of MDA-MB-231 cells. In addition, the murine solid Ehrlich’s adenocarcinoma growth and tumor metastasis were found to be inhibited by the treatment with Cuc A0-1, indicating that this triterpene could be a promising antitumor lead. Thus, this manuscript could be published in Marine Drugs as an Article. Only a few suggestions are shown below for the authors to refer.
1. The “anticancer action” shown in the title and/or the manuscript should be changed to “antitumor activity” to indicate the animal study.
2. Include a space before and after “±” in Tables 1‒3 (53.36±1.02 to 53.36 ± 1.02).
3. Discuss any molecular targets reported previously for Cuc A0-1 and Dj A.
4. Include references for triterpenoids and A2BAR if reported.
5. Discuss the relationships between SAR conclusions and docking profiles for Dj A.
Author Response
We are very grateful for the time and efforts you spent to evaluate our manuscript.
Q1: The “anticancer action” shown in the title and/or the manuscript should be changed to “antitumor activity” to indicate the animal study.
A: We thank the reviewer for his suggestion to improve our manuscript. Most of our work concerned the investigation of molecular mechanisms in TNBC cells in vitro, with only one experiment performed in mice. We would not change the title (if possible) as it more accurately describes our results.
Q2: Include a space before and after “±” in Tables 1‒3 (53.36±1.02 to 53.36 ± 1.02).
А: We thank the reviewer for the comment, we have corrected these shortcomings in the manuscript.
Q3: Discuss any molecular targets reported previously for Cuc A0-1 and Dj A.
А: These glycosides were isolated in our Institute, moreover, Dj A – is a novel compound, so they were not investigated earlier in relation to the action of any molecular targets.
Q4: Include references for triterpenoids and A2BAR if reported.
А: The interactions of triterpene glycosides and adenosine receptor are investigated now for the first time. Moreover, any data on the interaction of other triterpenoids and A2BARs are absent in literature.
Q5: Discuss the relationships between SAR conclusions and docking profiles for Dj A.
A: The following corrections were made in the section Results and Discussion and some data were added:
The SAR data showed negative correlation of the increasing of the quantity of hydroxyls mainly in the aglycones of the glycosides and in particular positions of their carbohydrate chains with their cytotoxicity. It was confirmed by the Dj D1 binding with the receptor/membrane system: the presence of the glucose in its chain as the second residue instead of the quinovose as well as the branching of the chain with fifth sugar unit dramatically influenced its binding with A2BAR. The presence of negatively influencing 23-OH group in the aglycone of Dj A and positively influencing free hydroxyl group at C-2 of quinovose residue in its linear sugar chain determined another mode of Dj A interaction with the receptor in comparison with Cuc A0-1 directly involving the sugar chain in the binding to A2BAR. The positive effect of methyl group of the quinovose was explained by the maintaining of such glycoside conformation that makes the electrostatic attraction of Dj A to allosteric site energetically favorable and the positive effect of methyl group of 3-O-methylglucose residue was due to the direct incorporation of this group into the binding site of A2BAR. In addition, the docking results improved with short-term MD simulation revealed the hydroxyl groups at C-2, C-4 of xylose residue as well as at C-4, C-6 of terminal residue are directly involved in Dj A interaction with functionally important receptor residues, responsible for the both modulation of several ARs subtypes by agonists and antagonists and for the ligand recognition by A2BAR type.
Reviewer 3 Report
Comments and Suggestions for Authors
Authors have presented the potential of sea cucumber triterpene glycosides cucumarioside A0-1 and djakonovioside A against human triple negative breast cancer. The concept of MS is good. However, the manuscript can be improved by improving the experimental section.
The specific comments, which could help to improve the manuscript are:
Manuscript should be revised for grammatical & punctuation errors.
Line 46: Write 685 thousand of women in millions to unify the given units in a sentence for better comparison.
Authors mentioned that the MD simulation (20 ns) studies were carried out. Explain the methods under method section, and present the results (The RMSD, RMSF, RG, interaction‐energy and SASA calculations to examine the biophysical movement of the ligand‐and‐protein complex).
Perform the in silico ADMET profile of both the triterpene glycosides.
As authors mention in title “Molecular mechanisms of anticancer action of sea cucumber triterpene glycosides”, it is better to illustrate the mechanism of action with the help of a figure.
Comments on the Quality of English Language
Moderate editing of English language is required.
Author Response
We are very grateful for the time and efforts you spent to evaluate our manuscript.
Answers to questions and comments in the attached file.

Reviewer 4 Report
Comments and Suggestions for Authors
In the manuscript (ID: marinedrugs-3187184) the studies were concerned to identification of the pathways triggered and regulated in MDA-MB-231 cells (triple-negative breast cancer cell line) by the glycosides, cucumarioside A0-1 (Cuc A0-1) and djakonovioside A (Dj A), isolated from the sea cucumber Cucumaria djakonovi. Different methods were used such as flow cytometry, fluorescence microscopy, immunoblotting and ELISA and QSAR.
In the manuscript the authors should take under consideration the following points:
a) The Figure 1 should be large because the structures of compounds are too small.
b) The quality of the Figure 2 is low.
c) How the QSAR model was build and how training set and test set of compounds were chosen?
d) Which descriptors were important in the QSAR studies?
Author Response
We are appreciated to Reviewer for the comments.
a) The Figure 1 should be large because the structures of compounds are too small.
Figure 1 containing better quality formulas was incorporated to the text and also provided as a separate file.
b) The quality of the Figure 2 is low.
Figure 2 containing better quality formulas was incorporated to the text and also provided as a separate file.
c) How the QSAR model was build and how training set and test set of compounds were chosen?
Model was constructed according to MOE-QuaSAR protocol of MOE 2020.0901 package. The main steps of the algorithms applied were added to the text (section 3.12. Building a QSAR Model).
For the training and tested sets, the compounds were divided into two groups balanced in terms of chemical diversity, and activity. 20 conformations for 13 glycosides were used as training set and 7 compounds including Cuc A0-1, Dj A and Dj D1 were applied as a test set. To build a QSAR model all currently available data on cytotoxicity of holothurian triterpene glycosides against MDA-MB-231 cells were used. Additionally, the Leave-One-Out cross-validation method was applied to the QSAR models obtained to mitigate possible overfitting risk of small size of dataset and provide a more accurate assessment of our model's performance.
d) Which descriptors were important in the QSAR studies?
Quantitative structure–activity relationships were calculated on the basis of correlational analysis of the physicochemical properties and structural features of 20 glycosides isolated from the sea cucumber C. djakonovi and their cytotoxic activity against MDA-MB-231 cells. In addition to the descriptors provided by the Calculate Descriptor tool of MOE CCG software (polarizability, refractive index, surface charge distribution, dipole moment, hydrogen bonds’ potential strength (donors and acceptors), hydrophobic volume, hydrophobic/hydrophilic areas, surface area, atomic valence connectivity index), following descriptors known to be significant for the activity of the glycosides (the presence/absence of 18(20)-lactone and the side chain, carbohydrate chain branching, and nature of the second sugar residue, the sulfate groups’ number and positions) were added to the descriptors training set.
QSAR results indicated the strong positive correlations between the availability of a normal non-shortened side chain, presence of 18(20)-lactone, and cytotoxic activity of the glycosides. The negative correlation of the molecular volume and shape was confirmed by higher activity of tetraosides with linear carbohydrate chains than corresponding pentaosides. Number and positions of sulfate groups were also among meaningful descriptors, but ambiguously affected on the activity of the glycosides depending on the architecture of their carbohydrate chains.
Only those descriptors that characterize the chemical features of the compounds djakonovioside A (Dj A), cucumarioside A0-1 (Cuc A0-1) and djakonovioside D1 (Dj D1) were included in the discussion. The most substantial correlations for the compounds activity were: the quantity of hydroxyls (reflected in such descriptors as: hydrophobic/hydrophilic areas and distribution, surface area, hydrogen bonds’ potential strength), carbohydrate chain branching (molecular volume and shape, principal moment of inertia), and nature of the second sugar residue (hydrophobic/hydrophilic areas and charge distribution).
According to the comments, the following information was added to the text:
Lines 436 – 448: For this purpose, 246 descriptors characterizing various structural elements, conformational and physicochemical characteristics, topological indexes, and functional groups were generated for 20 previously investigated glycosides [14, 15] isolated from the sea cucumber C. djakonovi. In addition to the descriptors provided by the Calculate Descriptors tool of MOE software (polarizability, refractive index, surface charge distribution, dipole moment, hydrogen bonds’ potential strength (donors and acceptors), topological indexes, describing molecular size and shape, hydrophobic volume, hydrophobic/hydrophilic areas, VdW surface area, atomic valence connectivity index), following descriptors known to be significant for the activity of the glycosides (the presence/absence of 18(20)-lactone and the side chain, carbohydrate chain branching, and nature of the second sugar residue, the sulfate groups’ number and positions) were added to the descriptors collection.
Lines 551 – 557: QSAR results indicated the strong positive correlations between the availability of a normal non-shortened side chain, presence of 18(20)-lactone, and cytotoxic activity of the glycosides. The negative correlation of the molecular volume and shape was confirmed by higher activity of tetraosides with linear carbohydrate chains than corresponding pentaosides. Number and positions of sulfate groups were also among meaningful descriptors, but ambiguously affected on the activity of the glycosides depending on the architecture of their carbohydrate chains
Lines 563 – 571: This was reflected in such descriptors as: hydrophobic/hydrophilic areas and distribution, surface area, hydrogen bonds’ potential strength. The finding was confirmed by the weak cytotoxicity of Dj D1 against MDA MB-231 cells because of the presence of glucose in its chain as the second residue instead of the quinovose, as well as the branching of the chain with the fifth sugar unit (hydrophobic/hydrophilic areas and charge distribution). The presence of a negatively influencing 23-OH group in the aglycone of Dj A was compensated by the linear tetrasaccharide chain with positively influencing free hydroxyl group at C-2 of quinovose residue (molecular volume and shape, principal moment of inertia).
Round 2
Reviewer 1 Report
Comments and Suggestions for Authors
The authors answered all question.
Author Response
We appreciated to Reviewer for the comments and thorough analysis of our research work.
Reviewer 3 Report
Comments and Suggestions for Authors
Authors have justified all the comments. However, graphical abstract is not provided.
Comments on the Quality of English Language
Moderate editing of English language is required.
Author Response
We are grateful to Reviewer for the comments and evaluation of our manuscript.
Graphical abstract was provided through Submission system. We are sorry the Reviewer haven't an access to this file. So, we attach it once again.
Moderate English editing was made.

Reviewer 4 Report
Comments and Suggestions for Authors
The manuscript was corected according rewiever suggestions and now is ready for publishing.